# New Power Structures and Shifted Governance Agendas Disrupting Climate Change Adaptation Developments in Kenya and Uganda

Julia Renner 

Department for Political Science, University of Koblenz-Landau, 76829 Landau, Germany;
rennerj@uni-landau.de

**Abstract:** Kenya and Uganda are currently two of the fastest growing countries in the East African Community. The political leaderships' prioritization of sociopolitical and economic development, combined with the wish for a closer integration into the world market, shifted the countries' governance structures and agenda setting. Undertaken economic projects, including oil explorations, mining and gold extractions, flower farming and intense rice growing, put conservation areas at great risk and led to a decrease of the country's wetland and forest cover. Accordingly, the impact of climate change on the vulnerability of countries is increasing. The paper critically investigates how particularly recent economic investments by national and international companies question the coherence between the institutional framework on climate policies, especially on a sub-national level of decision-making. Based on two field visits to the area, this paper raises the question of how the institutional frameworks shape climate governance processes in Kenya and Uganda. Looking at both political and climate governance structures from a pragmatic perspective, this paper concludes that the insufficient implementation of existing governance structures hampers the better integration of climate policies. National actors do not consider climate financing as an important issue which results in the fragmentation and undermining of climate policy processes.

**Keywords:** resource governance; economic upgrading; climate change adaptability

## 1. Introduction

In the past decade, climate change has become crucially significant worldwide. Many countries which are especially located in the climatic zones three (hot dry zone) and four (hot humid/tropical zone) are highly vulnerable to climate impacts, due to their geographical location [1]. Particularly in arid and semi-arid lands, this results, on the one hand, in persistent droughts and, on the other hand, it becomes more difficult to predict and to follow climatic seasons. Climate change, furthermore, amplifies stress on natural resources, especially on renewable resources, such as land, water or timber. Yet, this already leads to a reduction in crop productivity or resource degradations [2–4]. As middle income and least developed countries experience significant threats to livelihoods, assets and security [1], among the signatories, almost all developing countries committed themselves to undertake ambitious efforts to combat climate change and to adapt to its effects [5]. Governments in all countries play a critical role in enabling this transformation, which involves action from all aspects of society and the economy. To implement the climate goals, both functioning institutional structures and the provision of financial resources are needed to operate in a climate-neutral manner. Broadly, climate finance refers to local, national or transnational financing—drawn from public, private and alternative sources of financing—that seeks to support mitigation and adaptation actions that will address climate change [6]. To understand the importance of climate financing and the policy dynamics regarding

climate governance, a critical assessment of the signatory's environmental behavior is necessary during the implementation of the Paris Agreement. Furthermore, an assessment of the Nationally Determined Contributions to achieve these climate targets have to take place as well. One important aspect is to understand the institutional conditions and dynamics that influence how sub-national organizations respond to climate change. This includes organizations such as local governments and deconcentrated state agencies that operate at the 'meso-level' between the central state and communities, in the administrative spaces encompassing districts, municipalities and provinces [6,7]. From a strategic point of view, such organizations and agencies are by no means insignificant, and ideally, they can play a strong role in supporting households in climate adaptation. They are often responsible for implementing national climate change policies and interventions in practice, while at the same time being accountable to the local population. Simultaneously, their decisions about how to interpret and implement climate policies in practice have direct and often substantial impacts on livelihoods and the risks faced by climate-vulnerable people. Sub-national organizations inhabit an often opaque 'twilight' area between the central state and the community [7], where mandates and everyday actions are often open to interpretation and where public authority and 'reach' are sometimes fragmented, ambiguous and contested. Thus, they serve as a linkage between international and domestic institutions [8].

Kenya and Uganda are among the fastest growing countries in sub-Saharan Africa, and the East African Community in particular. Kenya is a country highly vulnerable to climate change, especially in its arid lands. Consequently, Kenya's national political commitment towards climate change mitigation issues is evident. For each economic investment undertaken, the Kenyan national government demands for an environmental impact assessment regarding the outcome of the economic investments. Similarly, especially the north of Uganda is highly vulnerable to climate change. Uganda therefore, committed itself to draft a national policy for disaster preparedness and management. For implementation, international cash flows are critical to support mitigation and adaptation actions. This is particularly critical for people who are most dependent on natural resources for their survival (e.g., farmers and pastoralists). To increase the countries' resilience to climate impacts, international actors (e.g., UNEP or the World Bank Group) have played a key role in supporting the development of climate change and disaster management policies of Kenya, through funding and the technical assistance of sub-national and local organizations [6]. All major developments in the institutional and policy framework around climate change have been funded and significantly been influenced by donors. In contrast, donor assistance in Uganda has accounted for a large share of the country's budget, but it is declining since 2003. However, most detrimental to the national political commitments are the economic development plans of both countries. In both countries, the percentage of the total budget towards climate governance constantly declined over the last decade. Even though the point of departure is similar in both countries, the institutional framework, and accordingly, the interplay between domestic and international institution regarding climate financing is somehow different. While in Kenya, devolution has been completed and thus political responsibility for climate and resource issues lies in the hands of sub-national and local institutions, Uganda is in the process of implementing decentralized decision-making processes. As it seems, recently, in Uganda climate change adaptation has become another resource in broader struggles over the authority between center and district.

In this mismatch, relatively recent, major economic activities started, which caused detrimental effects and have put climate governance and environmental impact assessment initiatives even further at risk. Studies show that the imbalance between resource availability, access and demand is increasing due to both natural and anthropogenic factors [9]. The natural and anthropogenic factors impacting on countries ecosystems and water basins include increasing threats from irrigated agriculture, water abstraction, population expansion, and economic projects, including oil explorations or mining. The operation of these commercial industries seems economically and socially important, however, it puts a great strain on the countries resources [10–14]. The aim to be better integrated into the world market and the desire to become middle-income countries in the next two decades resulted in a rapid

increase in economic infrastructural projects such as investments in the horticultural and floricultural industry, tourism facilities and other commercial activities impacting on the use and management of the countries land, water and other natural resources. Therefore, Kenya and Uganda's climate governance and financing are characterized by forms of durable patrimonial politics that provide some degree of accountability, but that also reinforce patterns of social exclusion.

The preceding discussion calls for a better understanding of how political and economic differences between the national and sub-national level of decision-making have hampered an inclusive natural resources and climate governance. A better understanding of the interactions and interests between various actors on different decision-making levels is important to explain the ambivalence between the commitment to implement the climate goals, the international funding received for better climate adaptability and insufficient institutional capacity. Most studies on the nexus of climate vulnerability and institutional effectiveness and resilience focus on areas which are already considered as "climate hot spots". Beyond those studies, existing studies either focus on the physical aspects of the country's institutional framework, the economic developments in general, or discuss the social impacts. Since the ambivalence between the effectiveness of the institutional framework and the economic development plans on the climate policy output is so great both at Lake Naivasha in Kenya and at Lake Wamala in Uganda, the two lakes serve as case studies for this research article. The two lakes are also equally suitable as case studies, as both areas are not yet so strongly influenced by climatic changes, and for this reason the ineffectiveness of proper climate governance must be explained primarily by other factors. The results drawn from these case studies can be transferred to other case studies across the East African Community and beyond, and therefore, general conclusions can be drawn.

The present article aims to address this divide, by assessing and comparing how the level of decentralization (institutional framework) impacts the climate policy output and linking it to its social implications. Therefore, the paper raises the question how the institutional frameworks shape natural resource and climate governance. The scope of this study has two principle objectives: (I) to identify the key characteristics of the institutional framework that shapes natural resource and climate governance. In addition, (II) the role and weight of internal decision-making processes and strategic priorities is discussed in shaping the overall climate policy agenda. To address these questions, field research was conducted in July and August 2018 and between July and October 2019, at both lake sites. The qualitative research was supplemented with government reports, existing literature on the economic agendas and climate policies. Using climate governance and the stakeholder analysis framework, this research article identifies and analyzes the main stakeholders living, working and influencing (climate) governance processes at the two lake sites, based on their interests, but also their power to influence both decision-making processes and the setting of national economic priorities. The paper provides a comprehensive understanding how insufficient governance structures, weak institutional arrangements and the insufficient cash flows hamper a better integration of environmental impact assessments of economic projects and climate financing thereof. Thus, the paper concludes that an insufficient implementation of existing governance structures hampers a better integration of climate policies, even though the countries still receive climate financing from international donors. Given that national actors do not consider climate financing as an important issue, the climate policy processes are fragmented and undermined.

The remaining paper is structured as follows. In Section 2, the theoretical frameworks of global (climate) governance and stakeholder analysis are described. Afterwards, the research areas, the methodology and data collection are introduced. In Section 4, a particular focus is placed on the institutional frameworks and the economic development agendas, which are crucial to classify the various actors regarding their interests and influences on climate policy at Lake Naivasha and Lake Wamala. In the final section, the findings will be discussed and summarized.

## 2. Theoretical Stakeholder and (Climate) Governance Framework

Decision making on sustainable natural resources management, economic investments and an inclusive climate governance requires the relevant knowledge to integrate environmental impact assessments, in regards to the economic developments undertaken. In the 1980s, Ostrom applied the previously used term polycentricity to outline governance structures that 'take each other into account in [a] competitive relationship, [in which] contractual and cooperative undertakings [ . . . ] have [ . . . ] central mechanisms to resolve conflicts' ([15], p. 831). Since Ostrom's publication, the debates and arguments have expanded, including aspects of efficiency and taking into consideration objectives in public administration. Independently from the development and exploration of the polycentric governance in diverse literatures, however, most scholars have drawn on the definitions used in the original conceptualization. Based on McGinnis and Ostrom, polycentric governance 'requires a complex combination of multiple levels and diverse types of organizations drawn from the public, private, and voluntary sectors that have overlapping realms' ([16], p. 15). However, in her concept, Ostrom focusses on a micro to macro-level action building and thus, does not fully grasp the importance of macro-level actors in decision-making processes. This is, nonetheless, is an important issue to consider when analyzing especially governance structures in developing countries, as otherwise, stakeholders are being blamed for decision-making problems who have not been in power to initiate policies in the first place. Discussing a web of relationships where actors and their environment are emergent and interpellated through their interactions requires one to watch the different levels of decision-making individually. All the same, Ostrom's concept does not show these power relationships and the connectedness between the different actors. Furthermore, Ostrom prioritizes individual users and local authorities over strong central governmental agencies, and hence, does not take in-between level governmental bodies into account. Lastly, Ostrom's concept fails to acknowledge the importance of the social power relationships and the inequalities which are potentially integral to them [15,16].

Accordingly, any decision-making process is not limited to formal governmental bodies anymore. Therefore, additional institutional features beyond the aspects presented in the core concept which are associated with achieving an effective climate policy output need to be integrated. Among other things, this includes multiple, and often overlapping actors, with different degrees of autonomy, to achieve the core attributes of a certain policy. What follows are acts of cooperation, competition, or conflict, to achieve a decision-making outcome between the identified overlapping actors. To devise a theoretical insight into the research objective stated earlier, thus, the paper proposes to combine the theoretical concept of climate governance, with the theoretical framework of stakeholder analysis (SA). The theoretical support for this proposition may be capable of exhibiting greater disparities between the level of decentralization and the climate policy output. Thus, it can provide insights into the overlapping claims various actors make regarding climate governance, and contributes to the understanding of the enabling or disenabling factors for a functional climate policy output and climate financing thereof.

Most countries of the world are grounded on complex institutional system, consisting of different types of institutions that allow people to work together and to achieve common tasks. The national government presides over these different types of institutions and people and can be characterized as an institution 'with rules about the responsibilities, rights and duties of certain elected or appointed officials and citizens' ([17], p. 23). Institutions can further be described by the task for which it was created and developed. Each institution provides governance arrangements, and should specify how decision-making, the allocation of resources and the how the measurement of performance should be undertaken. More so, they provide a set of governance arrangements, or 'system by which organizations are directed and controlled' [18]. The World Bank Group once indicated that for sustainable institutional effectiveness effective government organizations are required. Thereby, the governance arrangements should be defined in a way that all stakeholders and institutions should be able to participate in the decision-making process, according to the law applied, and that the decision should be made by broad consensus [19].

Likewise, the governance of environmental and climate issues is multifaceted, taking on a variety of forms and institutions [20]. Adger et al. differentiate three concepts of environmental governance. The first model represents a more traditional governance approach, looking at the interactions between the government and local stakeholders. The government is responsible for the formulation of policies and laws. An interaction or an exchange of ideas between the two levels is not taking place [21]. The continuation of the traditional model can be found in the co-management framework. The main responsibility for policy formulation and law promoting remains with the government, however, the responsibilities, rights and enforcement costs are shared between the government and the local actors. Given that appropriate governance structures are in place, more direct linkages between the agents of government (i.e., ministries, sub-national institutions) and the people on the ground are possible [22,23]. One development level further on, the actors develop linkages which complement governance processes. These linkages can appear horizontally as well as vertically. As local actors make common causes with other local actors or community groups, the linkage flows horizontally. Government agencies, similarly, share horizontal linkages with other political departments and ministries. Economic actors, scientific institutions or media share vertical linkages and often serve as linkage bodies between the government and the local stakeholders [24,25].

Thus, national governments are the first recipients who have to develop and integrate climate policies into their national policy processes. The country's legal system, furthermore, determines to what extent subnational institutions become an active player in either implementing climate policies, or considering environmental impact assessments [20]. In conclusion, climate governance happens both with and without the nation state, and can consequently be defined as a 'form of [ … ] political order [ … ] for a given political community on whatever level' ([26], p. 504). Taken together, climate governance involves multiple actors and sectors, with divergent interests and roles. This, in return, means that interests are more likely to be disaggregated around narrowly defined climate-related issues (e.g., carbon markets, economic upgrading or debt redemption) [20]. Climate governance demonstrates that climate-related policies are characterized by multiple sectors and various actors. Their interests and activities might lead undeniably to significant contentions between them, in case decision-makers fail to classify the various actor's perceptions, interests and influences.

Resulting thereof, governance takes place at various decision-making levels and involves multiple stakeholders. The engagement of a variety of stakeholders has the advantage to reduce enforcement coasts, as well as to increase the effectiveness and legitimacy of the governmental institutions and to finally enhance climate finance [14,21]. Thus, stakeholder analysis (SA), developed by Grimble and Wellard, is suitable for analyzing stakeholders' social action and structural characteristics, taking into consideration their interests and interactions in a given setting [27]. In particular, it proved to be successful in settings that are characterized by multiple stakeholders, multiple objectives and multiple interests [27–29]. Various scholars used the SA framework to analyze stakeholder's interactions from two points of view: their interests over a specific resource and their influence in terms of governance. Interests are 'based on [actor's] action orientation, adhered to by individuals or groups, and they designate the benefits the individual or group can receive from [a] certain object' ([14], p. 278). Neither interests nor influence can be measured directly. Therefore, a set of questions during the interviews was guided by Krott (2005) and Ogada et al. (2017). Krott suggests that 'interests are based on action orientation, adhered to by individuals or groups, and they designate the benefits the individual or group can receive from a certain object' ([30], p. 8). Similarly, influence was determined based on three aspects: the statutory role in resource management; the existing rights to the resources; and the extent of resources committed in resource management [14]. The purpose of the interviews during the field research was to first identify which stakeholders are relevant, with regard to natural resource use and climate policies. Second, the questions aimed to determine the stakeholder's interests with regard to natural resource use, but also their influence on climate legislation and thus its integration into national economic and environmental policies (see Section 3.3 on how the scores have been assigned).

Considering the stakeholders involved regarding their interests and influence, the core concept by Grimble and Wellard defined a four-group categorization classifying the actors into: "key players", "context settlers", "subjects" and "crowd" [28,31,32]. "Key Players" have both high interest and high influence and thus, they have an essential role in natural resource management and climate governance ([28], p. 5). Context Settlers are defined as those having low interest, despite being highly influential. Due to their importance in governance processes, they cannot be ignored, and hence, they should be monitored and managed [28]. "Subjects" have high interests but lack influence. Mostly, they are key to every process, even though they lack the ability to produce an impact. By definition, they are supportive. In order to increase their influence, they should form alliances with other stakeholders [28]. Actors in the "crowd" have neither influence nor interest. Therefore, there is little need to consider them in much detail, unless they are changing their interest or influence regarding the topic over time [28].

Based on the method of SA and taking into consideration the countries' governance structures, the stakeholders are classified in an "interest-influence" matrix. This matrix displays the groups' attributes and inter-relationships. The categorization of the various stakeholders into the four groups allows one to identify the consequences for climate policy processes arising from a particular institutional framework and the power of also non-state actors to influence national decision-making processes. The theoretical insights deviated allow to discuss any course of action (or non-action) and indicates an expected effectiveness (or, in the case of Kenya and Uganda, ineffectiveness) regarding climate policy output and climate financing.

## 3. Materials and Methods Used

### 3.1. Literature Review

In the past, a large part of the academic literature on climate finance has been concerned with the question of providing an accurate account of existing climate finance flows and discussing options to raise additional finances [33]. More recently, studies shifted their focus to identify areas towards which climate spending activities should be targeted. Thereby, the Green Climate Fund argued that each activity can take different foci, including cities, agriculture, forestry and energy, and can cover considerable mitigation potential, as well as adaptation needs [34–36]. However, it remains unclear how the most cost-efficient activities can be identified. In light of these shortcomings, several authors have argued for a broad and coherent strategy. In particular, climate finance is frequently regarded as an opportunity, not only to tackle climate-related aspects, but also to address sustainable development in a broader sense [33,37,38]. That is, climate finance is seen as a tool to achieve transformative change to redirect development patterns away from a lock-in of carbon-intensive energy infrastructures toward long-term green growth. However, efforts to achieve wide-ranging changes in economic structures by international donors run the risk of being perceived as interfering with other development objectives of the county's decision-making actors. Hence, it is of prime importance to confer 'ownership' of climate policies to recipient countries. Climate measures in developing countries are much more likely to succeed if they not only aim at reducing emissions, but also address additional domestic development objectives, such as ambient air quality, energy security, energy access, or public transportation [39–41].

Against this background, according to Carolina (2002), Lake Naivasha in Kenya and Lake Wamala in Uganda are of high economic and political importance [42]. The basin presents a wide variety of economic activities based around the water sources, with many different stakeholders often competing for the available resources. A study on turning conflicts into co-existence around the Lake Naivasha basin was carried out by Kioko (2016) [43]. He argued that various actors utilize diverse networks of relationships as adaptive responses to conflicts and socio-political dynamics at the local level. On the other hand, Ogada et al. (2017) reported that the better governance and managing of water resources require collaborative stakeholder networks within both basins. Further, they argue that even though both national and county governments and its agencies seem to command higher influence and interest

in resource management around the Lakes, the presence of influential and central stakeholders from non-governmental and private sectors play a key role in strengthening partnership in governance with varied interests [14]. Interactions in the basin are mainly guided by stakeholders' interest and sphere of influence, which have both promoted some level of participations in implementing a collaborative natural governance framework and to use international cash flow to implement the countries climate targets. A number of previous studies have discussed the potential impact of climate change on the availability of natural resources, and challenges resulting thereof. In particular, these studies focus on the most vulnerable in the countries. Mostly, the most vulnerable are defined as those who experience resource scarcities most times of the year (e.g., [44–52]). Other studies have also focused on how the influences of the international political economy and international regimes (e.g., international climate policies) impact on national governance processes and challenges with regard to their successful implementation of these policies on the ground [53–60].

However, taking into considerations the recently undertaken anthropogenic activities in both countries, the discussion intensified how climate financing in conjunction with environmental impact assessments are used effectively. The task should be to invest a specific amount of the available climate finance budget, so as to maximize beneficial results in climate- and other goal dimensions, where specifying a generalized rule for balancing the prioritization of different goals is difficult. However, for prioritization, diverse national climate policies require information that is often unavailable and susceptible to manipulation by powerful interest groups, which may arguably lead to the omission of cost-effective mitigation options. Complementary viewpoints may be needed to address climate finance failures [61–63]. Therefore, this study proposes that the point of departure for effective climate finance, the studies should first focus on the institutional effectiveness of the country (governance) and the different actor constellation influencing decision-making and climate financing internationally, sub-nationally and locally. This viewpoint might provide three main advantages. First, the interaction of national, sub-national and local institutions can be designed in such a way that an effective implementation of climate financing is possible, while at the same time taking national and regional interests into account; second, climate financing can be designed in a way that it becomes a simple, yet comprehensive way to increase and to channel economic investments; and third, climate finance could be used strategically to stabilize international climate policy cooperation, economic developments and an inclusive climate governance, especially with national and local institutions.

### 3.2. Research Area

Lake Naivasha is the second-largest freshwater lake in Kenya (see Figure 1), and its wetlands are a renowned Ramsar site of international importance [64]. Lake Naivasha is located in the Rift Valley Province, in Nakuru County. The Lake is furthermore covered by Naivasha District. Historically, Lake Naivasha basin was a Maasai grazing region and the main industry was fishing and farming, until 30 years ago. Over the last decade, the Lake basin has continued to experience a rapid increase in horticultural and floricultural industry, tourism facilities and other commercial activities which impact on the use and management of the Lake's water and natural resources [65–67]. The horticulture developments, tourism and industrial activities caused a tremendous shift in the landownership and population composition around the Lake. In summary, these factors resulted since the early 2000s in a sharp population increase in Naivasha town and its surroundings. The promise of labor opportunities has drawn many unskilled workers to the horticulture farms, the hotel industry and the power generation companies. At the beginning of the new century, the population of Naivasha and its surroundings was at around 300,000 [68]. However, according to recent estimates, there are nowadays between 1 million and 1.5 million people inhabiting the area [29,69,70].

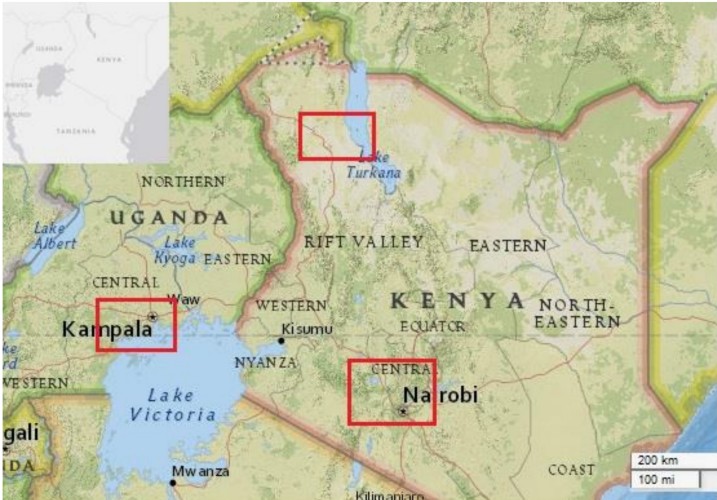

**Figure 1.** Research Area

Lake Wamala is located in central Uganda and is shared by the three districts Mubende, Gomba and Mityana. The Lake further falls under the jurisdiction of the Lake Albert water management zone. Historically, the Lake was directly connected to Lake Victoria and the main industry therefore was foremost farming and fishing [71]. However, due to its central location, its close proximity to Kampala and advanced infrastructural projects, the Lake is nowadays also of immediate concern for and of interest to the three districts, as administrative units as well as the national political units for economic projects. Resulting thereof, since the beginning of the century, the Lake has experienced an increase in overexploitation, degradation and bordering extinction. Contrary to Lake Naivasha, Lake Wamala basin did not experience a sharp population increase. Instead, the population around the Lake decreased, as land was either taken over by companies or was degraded and, resulting thereof, became unusable for agricultural purposes. Only the three fishing villages Lubajja, Lusaalira and Buzi Bazi were preserved [20,71].

The operation of these commercial industries at both lakes seems economically and socially important in this semi-arid environment, but put great strain on both basins shared land and freshwater resources [10–13]. Both regions experience bimodal rainfall patterns. However, its frequency and duration vary due to the effects of climate variability and change considerably, resulting in substantially changes of Lake Naivasha's and Lake Wamala's water levels over the last 50 years [72]. Whereas Lake Wamala experienced a tremendous shrinking of its lake levels in the early 1990s, Lake Naivasha reduced by one-third of its size in 2009, as a result of a major drought. Afterwards, the water levels of both lakes increased, which led to Lake Naivasha recording the highest water levels in its history in 2018 [72,73]. Contrastingly, Lake Wamala did not recover its water levels to its original size. In summary, both areas are under constraint from both natural and anthropogenic factors.

*3.3. Methodology and Data Collection*

This study collected both qualitative and quantitative data, through various approaches. Qualitative and interpretative social research combines multiple forms of interviews, and field research was used to describe the research object from the perspective of the 'actors in everyday contexts [ . . . ] and to study complex social actions and practices of everyday situations [74]. In addition to the primary data obtained from the interviews, this work is also based on the evaluation of quantitative data.

Applying the data collection to the theoretical framework of stakeholder analysis and (Climate) Governance, the researcher followed a five step approach to analyze the received data [75]: (I) stakeholders involved in the resource sector were identified and how they influence or impact

on the environmental situation, (II) the stakeholders have been categorized; (III) information was collected via interviews and observations and analyzed; (IV) the relationship between the stakeholders were investigated and put into an "interest-influence" matrix; and (V) a cause-effect relationship was developed, to explain why climate governance and climate financing is not working, despite efforts to combat climate change. The separate analysis of the data retrieved from the fieldwork in Kenya and Uganda served as a cross-check of the data received and the conclusions drawn.

Prior to the fieldworks, extensive literature reviews on climate governance, stakeholder analysis, the national economic development plans, as well as natural resource governance structures in both Kenya and Uganda, were conducted. Thereafter, participatory research at various locations in the countries took place. Furthermore, interviews have been conducted and formed the main method for data collection. The primary data used during the research process has been obtained through semi-structured and structured interviews with different groups of actors, from experts to representatives of national and sub-national political and economic organizations to local resource users. Besides expert interviews, the field research also involved focus group discussions with local resource users in remote parts of the countries. Contrary to the expert interviews, the community interviews have not been structured and were conducted in Swahili and Luganda, and translated into English by a local research assistant simultaneously. This open method gave the community members the opportunity to talk about issues they find most pressing, as they threaten their everyday survival.

In total, more than 50 interviews have been conducted. The interviews took place in July and August 2018 and again in summer 2019. Interviews with international and national stakeholders were conducted once. Interviews with local resource users were conducted during both research visits. The individual and small group interviews were based on an interview guide structured along the themes of 'institutional framework', 'general changes', 'level of interest in sustainable climate policy', and 'influence to impact sustainable climate policy'. Whereas the questions concerning the first two aspects have been open, the interviewees were asked to rank their level of interest and influence as either high interest/influence or low interest/influence. Based on this classification, the stakeholders have been categorized in the four-group categorization.

Other crucial data sources embraced interviews with flower farm and industry/company representatives and researchers, but also 'informal interviews' with consultants and development organizations during coffee or dinner appointments. The used approaches were helpful for one to understand the complexity of resource governance dynamics and the conflictual interests regarding climate governance thereof. The data collected from within the natural resources and water sector were supplemented or triangulated with sources of information from outside the resource industry and included e.g., international organizations, national political and economic companies, public authorities and non-governmental organizations (NGOs).

## 4. Results

The result section is structured along the two research objectives. In the first part, the key characteristics of the national institutional frameworks are described, followed by a listing of the key stakeholders involved in climate policy at various levels of decision-making. Tables 1 and 2 present the identified stakeholders, who have both interest and influence in climate policy. The second part of the section addressed the role and weight of the stakeholders' interest and influences on climate policy output, in relation to the institutional framework and the economic development agenda. To determine which stakeholders are involved in climate policies and climate decision-making, the participants were asked to name all stakeholders they consider to be relevant (see Appendix A). The interview questions for the stakeholder classification were as follows: (I) Which of the named actors do you consider to have high interest in climate policies? Which actors have low interests in climate policies? (II) Which of the named actors do you consider to have high influence on the institutional framework? Which actors have low influence on the institutional framework? In order to identify the classification, all interview participants were only allowed to categorize the actors in the groups of "high" and "low".

**Table 1.** Stakeholder Classification Lake Naivasha.

| Political Actors (National) | Economic Actors (International) |
|---|---|
| *Governmental Agencies* | 27. International Flower Farms (Dutch and British) |
| 1. National Environmental Management Authority | 28. International Investors (e.g. Chinese and Japanese) |
| 2. Water Resource Authority | |
| 3. Water Tribunal | Economic Actors (national) |
| 4. National Water Storage Authority | 29. Hotels |
| *Governmental Ministries and Departments* | 30. Geo-thermal industries |
| 5. Government of Kenya | 31. Power Plants |
| 6. Ministry of Water and Sanitation | 32. KenGen |
| 7. Department for Defense | |
| 8. The National Treasury | Resource Users (local) |
| 9. Ministry of Energy | 33. Pastoralist Groups |
| 10. Ministry of Devolution and the Arid and Semi-Arid Lands | 34. Fishing Community |
| 11. Ministry of Lands and Physical Planning | 35. Farming Community |
| 12. Ministry of Agriculture, Livestock, Fishery and Irrigation | 36. Villagers at Lake Naivasha and in the upper catchments |
| | |
| Political Actors (sub-nation/county-level) | NGOs and community-based organizations (local) |
| *Governmental Agencies* | 37. Lake Naivasha Riparian Association |
| 13. Basin Water Resource Committee | 38. Lake Naivasha Resource Users Association |
| 14 Water Harvesting and Storage Authority | 39. Community Forest Association |
| 15. Water Storage Board | 40. Imarisha |
| 16. Lake Naivasha Resource Authority | 41. Lake Naivasha Flower Council |
| 17. Water Sector Trust Fund | 42. Kenyan Wildlife Service |
| 18. Naivasha Water and Sewage Cooperation | 43. Beach Management Unit |
| 19. Water Services Regulatory Board | |
| 20. The National Water Conservation and Pipeline Corporation | I(N)GOs/Agencies (International) |
| 21. Regional Water Services Boards | 44. World Wildlife Fund |
| 22. Water Works Development Agencies | 45. World Bank |
| 23. Water Basin Committee | |
| | |
| *Governmental Ministries and Departments* | |
| 24. County Government Nakuru | |
| 25. Ministry for Environment of Nakuru County | |
| 26. Department for Water within the County Ministry for Environment | |

**Table 2.** Stakeholder Classification Lake Wamala.

| Political Actors (National) | Economic Actors (National) |
|---|---|
| *Governmental Agencies* | 24. Ugandan Breweries |
| 1. National Environmental Management Authority | 25. Sand Miners |
| 2. National Water and Sewage Cooperation | 26. Gold and Mineral Miners |
| 3. Umbrella Organizations for Water Provision | |
| 4. Petroleum Authority Uganda | Resource Users (local) |
| 5. Investment Authority | 27. Villagers/Community |
| Governmental Ministries and Departments | 28. Farmers |
| 6. Government of Uganda | 29. Fishermen |
| 7. Ministry of Finance | 30 Pastoralists |
| 8. Ministry of Energy and Mineral Development | |
| 9. Ministry of Water and Environment | NGOs and community-based organizations (local) |
| 10. Ministry of Land, Housing and Urban Development | |
| 11. Ministry of Works and Transport | 31. Kikandwa Environmental Association |
| 12. Ministry of Agriculture | 32. Rural Development Media and Communication |
| | 33. Uganda Coalition for Sustainable Development |
| 13. Parliament | 34. National Association of Professional Environmentalists |
| 14. Natural Resources Management Group of Parliament | 35. Wetland International |
| | 36. Ecological Christian Association |
| | 37. Civic Response on Environment and Development |
| Political Actors (sub-nation/county-level) | |
| *Governmental Agencies* | I(N)GOs/Agencies (International) |
| | 38. Gesellschaft für Internationale Zusammenarbeit (GIZ) |
| *Governmental Ministries and Departments* | 39. Kreditanstalt für Wiederaufbau (KfW) |
| 15. Environmental Officer Mityana County | 40. French Development Agency |
| 16. Environmental Officer Mubende County | 41. Italian Agency for Development Cooperation |
| 17. Environmental Officer Gomba County ( | 42. European Union |
| 18. Lake Albert Water Management Zone Officer | 43. World Bank |
| 19. Forestry District Officer | 44. World Wildlife Fund for Nature |
| 20. Local District Officer | 45. International Committee of the Red Cross |
| | 46. Oxfam |
| Economic Actors (International) | |
| 21. Coca-Cola | |
| 22. Total | |
| 23. Chinese Companies/Investors | |

## 4.1. Governance Structures in Kenya and Uganda

Kenya has embraced a decentralized institutional framework (devolution according to the Kenyan legislation) for the almost all decision-making processes, including natural resource development and climate policy that emphasizes multi-stakeholder participation. With the successful completion of devolution in Kenya in 2013, both the planning and implementation of natural resources related policies devolved to the counties. Therefore, the counties founded ministries which shall be responsible for the development, maintenance and management of the county's natural resources and a related environmental impact assessment of economic developments. However, the overall responsibility for natural resource management and climate governance remains with the respective ministries in Nairobi, as every natural resource 'is vested in and held by the national government in trust for the people of Kenya' ([76], p. 5). Whereas the national government and the Ministry, respectively, is the owner of the individual natural resources, the regulation, the management and the use of these resources is provided through an authority which serves as an agent on behalf of the national government ([76], p. 6). These governmental agencies have, among other tasks, four main duties: (I) to formulate and enforce standards, procedures and regulations for the management and use of natural resources;

(II) to receive and to issue permits; (III) to set and to collect fees, and (IV) to formulate policies on the development of natural resources ([76], p. 12). Due to its decentralized structures, the country was further divided into five larger drainage areas, each responsible for a specific basin area and the management and maintenance of the natural resources ([76], p. 25). Each drainage basin is subdivided again into committees. Their primary task is to achieve a wider stakeholder participation and to act as an advisory to sub-national and local Natural Resource User Associations. However, each of them shall operate under the regulations made by the governmental agencies ([76], pp. 26,27,29). Even though the constitution delegates extensive authority to political sub-national and local level service providers, these are unable to function without adequate financial support from the Kenyan Ministry for Finance, called The National Treasury. In the last Budget Statement for the Fiscal Year 2018-19, The National Treasury allocated 3.01 percent of the total governmental budget to the environmental, water and natural resource sector [77]. In 2018, 51 Million people lived in Kenya. The expenses for water resources amounted to 13 USD per capita.

Since 1993, Uganda is undergoing a decentralization, which is shifting national governmental responsibilities to the district governmental level. After 2009, the districts started to become the main implementers for natural resource governance issues [78]. However, the legal framework is still guided by the 1997 Water Act and the 2001 developed Land Act. The laws came into force when the institutional framework for natural resources was still centralized and therefore governed from Kampala. This overlap of the institutional frameworks is exemplified, e.g., in the article about the ownership over the country's water bodies. All rights to water in Uganda are 'controll[ed], protect[ed] and manage[d]' by the government and 'exercised by the Minister and the director in accordance with this Part of the Act' ([79], p. 5). The Ugandan government defines that the overall objective of the policy is 'to manage the [ … ] resources in ways that are sustainable and most beneficial to the people of Uganda'. Despite the decentralized structures, the "key player" in natural resource governance is the central government through the Ministries. Each Ministry is responsible for defining the national directives and standards, for setting development priorities and for managing the resources. It furthermore monitors and evaluates the development programs and to keep track of the sectors efficiency and effectiveness [80].

In the same way as Kenya, Uganda also introduced a sector reform and is in the process of establishing four sub-national water management zones, which are responsible for the management of all resources within its zone. Each of these four management zones are composed of respective catchment management organizations, which shall develop detailed catchment management plans for the sub-catchments under their jurisdiction. In addition, each management zone is subdivided into sub-catchments, in order to ensure a better implementation of the mentioned policies down to the local level. On average, three to four districts form a sub-catchment [81]. Compared to the pre-reform state, this structure is much more differentiated, because it demonstrates an improvement in terms of policy performance, the issuing of permits, monitoring and the enforcement of laws, compliance and regulations [80]. Each management zone consists of several sub-catchments, which furthermore form micro-catchments. District offices are responsible for managing the resources of each micro-catchment at the district level. Similar to Kenya, the management zones and the sub-catchments, as well as the districts, cannot perform their respective duties without sufficient financial support. In the last budget statement for the fiscal year 2018-19, the Ministry of Finance, Planning and Economic Development allocated 5.04 percent of the total governmental budget to the water and environmental sector [82]. As of 2019, 42 million people lived in Uganda [83,84]. Resulting thereof, the government allocated 8 USD per capita to the water and environmental sector. Compared to European countries, both countries allocate comparatively little to the environmental and natural resource sector and therewith spend less on climate financing. Compared to the United States, however, both countries spend slightly more. In 2018, Germany allocated 36 USD per capita to the water and environmental sector [85,86]. France allocated 204.19 USD per capita to the environmental and sustainable sector [87], and the United States spent 2.58 USD per capita on environmental programs in 2019 [88].

### 4.2. Description of the Current Actors in Kenya and Uganda

As highlighted previously, Lake Naivasha is at a short distance from Nairobi, and therefore close to Kenya's major international airport. This geographical proximity came to be appreciated by both political and economic actors. After independence, the national government promoted Lake Naivasha's picturesque setting and advertised the Lake's aquatic bird diversity, as well as wildlife density and, therefore, the Lake became both popular for international tourists and residents of Nairobi as a weekend escape [89]. Furthermore, the national political elite made use of the permeable and fertile soils found around the Lake's shorelines and sold or rented land to entrepreneurs and international commercial flower companies. Since the late 1980s, national and international business companies moved in which at first mainly cultivated flowers for export. Later, at the beginning of the new century, other economic companies rented or bought land to set up geo-thermal and hydro-power companies [90,91]. The horticulture developments, tourism and industrial activities caused a tremendous shift in the landownership and population composition around the Lake. The space and resources around the Lake are nowadays shared by three broader groups: local resource users like farmers, pastoralists, fishermen and villagers, and national and sub-national political actors and governmental agencies, as well as multi-national companies and national hotel owners (Table 1).

Likewise, the population composition around Lake Wamala changed. Whereas Lake Naivasha experienced a tremendous population increase due to numerous economic activities, the increase in economic activities led to a decline in the population living at Lake Wamala directly. One of the reasons for this decline was the increase in sand, gold and mining activities around the Lake, as well as deforestations, land and wetland degradations, to increase the space for agricultural activities such as rice growing. These activities are partly carried out by national and international companies, which are granted land rights through the national government to pursue their business. Being originally characterized by fishing, local resource users (farmer, fishermen and villagers) and multi-national companies, as well as (sub)national actors and (inter)national non-governmental organizations, share the space in the wider surroundings of the Lake (Table 2).

Neither the Kenyan nor the Ugandan national government is interested in a sustainable climate policy at Lake Naivasha and Lake Wamala. Rather, both governments' interest is driven by an economic development of the area to achieve a middle-income status by 2030 (Kenya) and 2040 (Uganda) as laid out in the economic development plans (see Section 4.4). Thus, the governmental actors and its agencies promote economic infrastructural projects to increase the countries' GDPs, through promoting the tourism sector and the export of flowers (Kenya), as well as the exploration of sand and other minerals and the farming of agricultural products, like rice (Uganda). Independently of finalizing devolution, Kenyan national agencies are unable to manage climate related policies without support from the government and the respective ministry, thus, their interest in pushing climate policies on the agenda is low. In Uganda, decentralization is still ongoing and most agencies are in the process of being set up. Accordingly, they are unable to voice their interest in climate-related policies, as they lack both the man-power and administrative structures to work autonomously.

Kenyan sub-national ministries, departments and agencies mentioned interests concerning better climate policies and a sustainable handling of the environment and its resources. They are foremost concerned about a better regulation and the management of the natural resources to limit the impacts of climate change. Furthermore, they are interested in integrating improved conservation and the sustainable use of resources to limit the environmental vulnerability of the area. More so, county officers are interested in the sustainable development and management of natural resources to counter climate-related effects. They highlighted, however, that they do not have the capacities to implement these interests, as they lack the funding and power to implement them. As they cannot act apart from the national government, they are interested in keeping close ties with the national authorities and therefore, promoting the economic development of the area. Respectively, the economic development interests outweigh their interest in climate policies. In Uganda, sub-national ministries and departments argued

that, even though they would like to be interested in climate policy and sustainable environmental policies, they are foremost interested in establishing sub-national structures.

Local resource users voiced interest in climate policies, as they are the most dependent on the natural resources around the two lake sites. Hence, their interest in climate policies is driven by different motives regarding the land and water resources around the Lakes. Pastoralists use land and water for watering and grazing their cattle, farmers need land to pursue their agricultural business and to water the fields. Fishermen are interested in free access routes to the Lake's shorelines and villagers need water and land for everyday life activities, including e.g., cooking, drinking, or washing.

During the interviews, neither international nor national economic actors expressed direct interest in climate policies. They rather argued that they recognize that climate policies are an integral part of their corporate sustainability agenda. Thus, climate policies are part of any other interests the economic actors pursue. They, furthermore, indicated that they are interested in a preferential access to the Lake's resources to pursue their economic businesses. Finally, non-governmental national and international organizations have an interest in climate policy. Accordingly, they are interested in sustainable management of the resources, the improvement of the local resource users' livelihood and therewith, the empowerment of local resource users. Moreover, some community-based organizations are also interested in using and managing the resources in a way to preserve environmental stability, as well as to foster the social-economic development of the area. In particular, international organizations are interested in supporting national stakeholders, to ensure the sustainability of economic projects and political infrastructure projects, through the provision of financial means. Furthermore, non-governmental organizations expressed criticism that the national government is using the money to promote their economic agendas through its ministries (see Section 4.3), instead of advancing the integration of climate policies in the institutional framework.

*4.3. National Economic Transformation*

Kenya has committed itself to facilitate economic development, to achieve a middle-income status by 2030. The country aims to obtain a medium-term domestic product of 6.0 percent [92]. In order to realize this vision, Kenya has set up the Big 4 development agenda [93]. To achieve this ambitious goal, targets have been established in four sectors. The Big 4 agenda will prioritize manufacturing, universal healthcare, affordable housing and food security, as well as social development. To achieve these goals, Kenya has already invested 20 percent of its GDP in manufacturing, agrobusinesses and services as well as 12.7 percent of its GDP in infrastructural projects [92]. As of the end of August 2019, Kenya furthermore shifted its focus to support three major projects in the energy sector. The largest of these three projects is the construction of an oil pipeline from Turkana to Mombasa [92]. The estimated costs are roughly $1 billion. Another large infrastructural project is the construction of a cargo center at Lake Naivasha. Whereas the oil pipeline is financially supported by Chinese investors, the World Bank Group is the main financial contributor for the cargo center at Lake Naivasha [69,70,94].

Uganda's national economic development strives to reach upper middle-income status by 2040. The second five-year National Development Plan (NDPII) and the president's strategic priorities for 2016–2021 [95] stress the importance of focusing on macroeconomic stability and investing in health and education, as well as the creation of strategic agricultural land, urbanization and transportation [96]. To achieve these targets by the end of the five-year plan, four core elements have been formulated, which include an increase in sustainable production and productivity, an increase in strategic infrastructure to accelerate Uganda's competition and to enhance human capital development, as well as to strengthen mechanisms for quality, effective and efficient service delivery [95]. After the successful realization of these targets, the national economic development plan identified five more growth drivers; namely agriculture, tourism, minerals, oil and gas, infrastructure, and human capital development [95].

To achieve the set goals, in both countries, emphasis has been given, to ensure rapid growth through deforestation or wetland degradations. The vacant areas are used for, among other things,

low-income housing estates, or as building plots for national and international companies. At Lake Naivasha, the government aims to increase the area of cultivable land for horticulture, hydro-power companies or hotels. Thus, this policy necessitates the conversion of forestland, national parks and shorelines into cropland or building plots. Around Lake Wamala, most wetlands have been degraded. Nowadays, they are used for extensive rice-growing, mining and the set-up of housing areas. Additionally, to gain high productivity rates in agri- and horticulture, fertilizers, pesticides and other chemicals are used, as it is expected that they increase the agricultural productivity. However, this in return increases the amount of nutrients such as nitrogen, saline, mercuries or phosphorous in the Lake and its surroundings. Despite both countries' political agenda to include environmental impact assessments with regard to economic developments undertaken, these infrastructural projects and the agenda put forward with the national economic plans increase the pressure on habitat and spawning grounds and thus, hampers sustainable climate governance. Since the turn of the millennium, the wetland-cover of Uganda decreased from 15 to 9 percent and the national forest cover stands, nowadays, at 23 percent [97].

Resulting thereof, the projected economic growth supported by the current national policies embedded in the institutional framework is accelerating the climate change vulnerability of the countries, especially in areas which are not yet considered as climate hotspots. Moreover, the current trends show that international climate financing is used for economic development projects, instead of supporting sustainable approaches to limit the effects of climate change in both areas.

### 4.4. The Interaction of Interests and Influences

The role of international finance to produce a climate policy outcome is mainly determined by the institutional framework and the capacities the stakeholders have to influence climate decision-making processes. Figures 2 and 3 summarize the position of the identified stakeholders on their interest in climate policy and their influence to integrate climate policies in the national decision-making processes regarding investments undertaken. The classification is based on the conducted interviewees with the stakeholders.

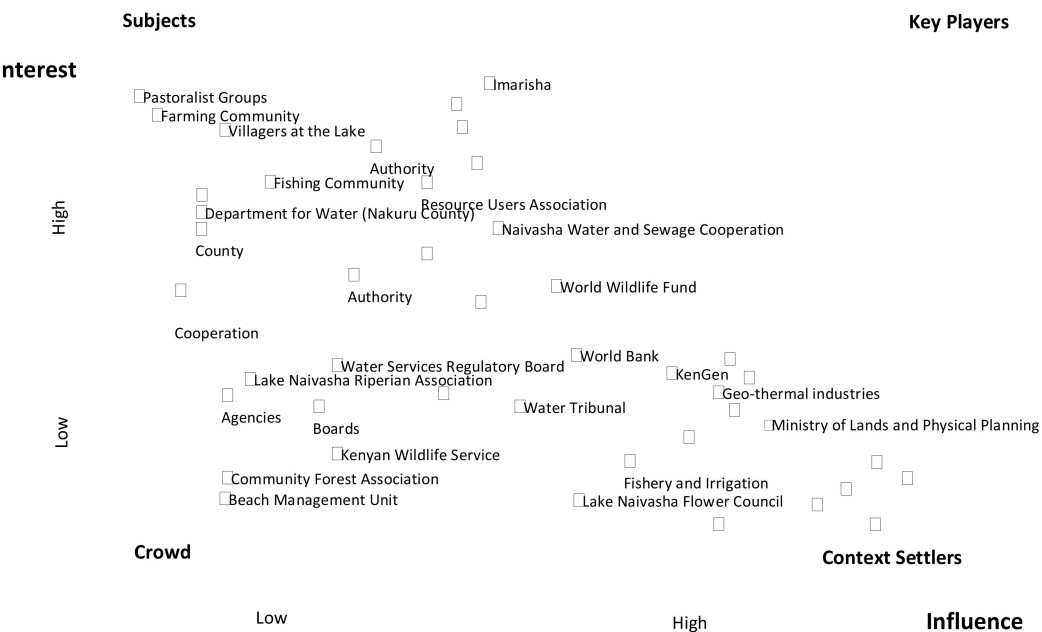

**Figure 2.** Influence-Interest Matrix Lake Naivasha.

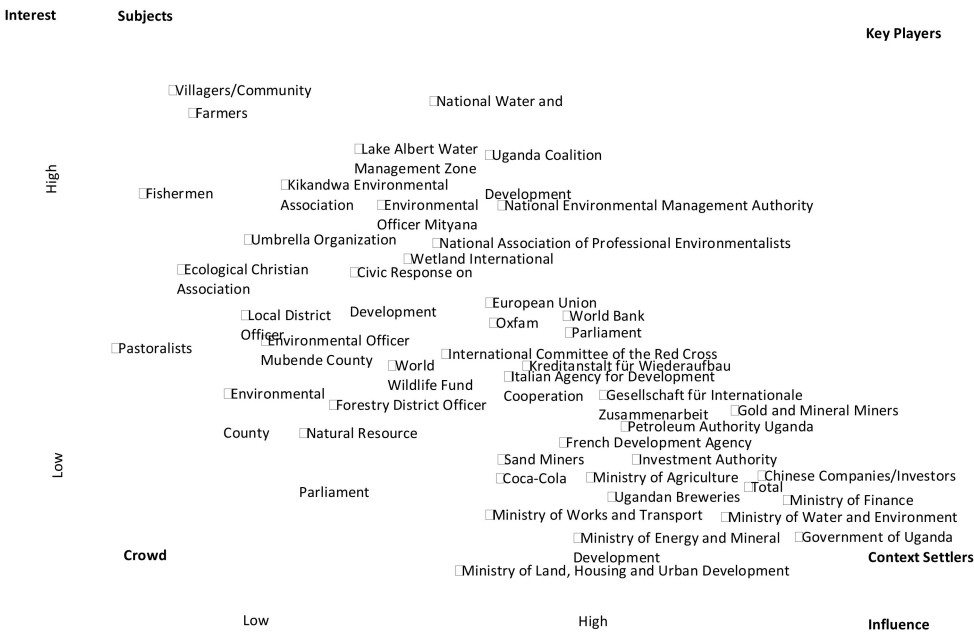

**Figure 3.** Influence-Interest Matrix Lake Wamala.

As shown in Figures 2 and 3, the majority of stakeholders are in between "subjects" and "crowd", however, the majority lean more towards "crowd", with high interest but little influence on climate policies. Across both countries, there is a total of 10 "context players" with low interest in climate policy, but high influence on the institutional framework. The governmental agencies/ministries, and international economic actors have the highest influence on the policy setting in the Lake basins. Government agencies/ministries have the lowest values for interests, followed by the business sector, the international and national (non)governmental organizations and the resource users.

The national political actors' low interest climate policies originate from the economic development agendas, which are overshadowing other policies. Their high influence on decision-making processes is grounded on the national legislations that state that 'every [natural] resource is vested in the state' [76,77], and therefore the government and its units have a mandate to manage and protect it on behalf of the citizens. Furthermore, the legislations grant the governmental units the regulatory powers and authority, which they exercise in managing the basin.

International and national economic actors are classified as "context players" as well. The business sectors' high influence arises out of the national government's focus on the economic development agendas. As such, the economic stakeholders get preferential access to the Lake's resources from the national actors. Furthermore, their high influence on the institutional framework results from the national governments dependency on them, as the governments are unable to achieve their economic targets without the economic actors' generosity to use both Lake Naivasha and Lake Wamala to set up their sites. Furthermore, economic actors argue that they improve the livelihood of the local resource users by offering labor to the villagers (flower farming and artisanal mining), and therewith support the national government's claim to increase the employment opportunities for especially unskilled workers. Thus, despite the low interest of the economic companies in national climate policies, they are highly influential in guiding the institutional context of Kenya and Uganda.

In Kenya, sub-national departments and ministries are classified in between "subject" and "crowd". According to the existing legislation, their influence in integrating climate policies in the institutional framework should be high. They are charged with the supervision and coordination of all matters related to environmental, natural resource and climate governance. This mandate shall give them greater influence in managing all political affairs in Lake Naivasha basin and its respective resources. However, their influence is still low, as they lack the necessary structures, financial means and strategies

for sustainable resource management. Moreover, their interest in the demand for the implementation of the existing institutional framework is low, as political kinship, ethnic affiliations and patronage along political lines are widespread problems in Kenya and also affect sub-national ministries and departments (rank 137 of 180 countries). Compared to Kenya, in Uganda, sub-national ministries and departments are first and foremost located in the "crowd" with low influence and low interest. As Uganda is still in the process of decentralizing its institutional framework, sub-national departments which shall deal with climate and environmental policies are established currently. Employees of the newly established sub-national departments articulated the importance of climate governance, however, they stressed that their primary interest is both in the set-up of administrative structures and ensuring financial means. Additionally, the employees are also motivated to secure their positions in the departments and ministries, and consequently, they mainly represent the interests of the national actors in their areas of influence.

Besides, local NGOs and international NGOs have the lowest influence, despite their high interests in integrating climate policies in the institutional framework. Most likely, their low influence is due to the diversity of the NGOs engagements (thus reducing their focus). However, the World Wildlife Fund and Imarisha at Lake Naivasha and Wetland International and Uganda Coalition for Sustainable Development at Lake Wamala are outliers, due to their higher levels of influence compared to the other international NGOs. All four have been actively involved in the basins for over 15 years, focusing on catchment protection and resource users' livelihood. Imarisha and Wetland International have further been successful to get parliamentarian groups involved in their work, and to set environmental and climate policies on the political agenda, albeit, so far without success on a parliamentarian hearing, or even a legislative resolution. Thus, the attempts by NGOs to acquire control over political actions concerning environmental management processes have, however, failed. Nevertheless, they are still interested in climate policies and the sustainable management of resources. Their aim is to support the local actors in resource access and to increase the production of crops and generate income from the resource use for the local actors. The local resource users rely on the basin's resources to manage a sustainable way of life, however, their institutional influence is close to zero and, therefore, is not considered further in detail.

The results from the SA indicate that a variety of stakeholders is involved in the Lake basins, each with different levels of interest on the implementation of climate policies. The analysis revealed that the level of interest in integrating climate policies and the level of influence on the institutional framework are contradictory. Accordingly, in the mid-term, climate policy will likely be overshadowed by the national economic agendas. More so, decentralization and devolution, whether still ongoing or completed, will not shift governmental responsibilities to the sub-national level of decision-making. Thus, sub-national actors who seem to have a slight interest in climate policy do not have the power to exercise influence, given the current institutional context. Moreover, they do not have the opportunity to access international climate finance and therewith are unable to produce any climate related policy outcome. National political actors use their influence to determine the direction of political guidelines in the basin. Thus, they use the received international cash flow to invest in infrastructure and other economic projects, to accomplish the targets set out in the economic agendas. Correspondingly, the sums have not made any difference on the ground. However, apart from a few boreholes drilled along the roads, the process to integrate environmental impact assessments regarding the economic investments undertaken has been slow and the impact is limited. Overall, the results show that the question of integrating adequate climate finance in the institutional framework is answered through the interplay of national and sub-national actors, despite other actors' interest in the Lake and the projects undertaken. Taken together, climate policy and climate finance might become a resource in the broader struggles over the authority of decision-making processed between center and district.

## 5. Discussion

The national economic agendas increase the climate change vulnerability of the area, which in turn increases the demand for climate financing, to limit the Lake basin's exposure to climate change. For instance, the damming of the incoming rivers, extensive sand and mineral mining or the treatment of flowers with chemicals are detrimental to the environmental impact assessments. Moreover, the cultivation of agricultural land with rice, or the planting of exotic species that are well suited for export negatively affect the habitat and lead to degradations, the drying-up of the Lake's shorelines and, therefore, also negatively affect the nursery sites of the fish species in the Lake. As such, effective institutional frameworks are required, to ensure the integration of climate policies in the institutional frameworks. However, shared state authority between national and sub-national political authorities over environmental issues, and therewith climate finance, is not a given in both countries. In Kenya and Uganda, the state and its functions are well established centrally, but in practice have incomplete 'reach' on the ground. The state sees its legitimacy over decision-making processes being challenged by especially sub-national institutions.

In Kenya, devolution provides an institutional framework for tapping into diverse climate funding sources, especially enabling a cash flow from the global to the national level and further down to the sub-national ministries to implement climate policies. However, internal mechanisms and weak synergies between various policies and institutions led to a high degree of fragmentation and contestation of the governance architecture. There were indications that devolution could affect sub-national policies positively. However, ethnic affiliations and political kinship have made sure that leadership position on the county level has been occupied with people from the same ethnic group, to ensure electoral success, but also the implementation of national policies in the counties. Interviewees in sub-national departments reported that the initial excitement of devolution and better policy implementations has already been replaced by frustration and led to resignation.

Uganda is decentralizing its institutional framework; sub-national departments have been established, however, most of them lack the equipment and financial resource to start operating. Whereas the state sets up policies fitting the national economic agenda, the central state takes up an ambivalent position in matters of climate policies. So far, mandates and resources addressing climate disasters have been exempted from decentralization. This, in turn, reflects the interest of the central government and its respective ministries to maintain control over environmental responses and received climate finances. The governance architecture in Uganda is fragmented, and results in a loose network of various institutions, reflecting national decision-making processes and policy priorities. Resulting thereof, sub-national institutions do not expect to receive decision-making authority from the national government in the mid-term.

The analysis has shown that there does not exist any decision-making authority that is clearly responsible for climate policies, as well as policy formulation, regulation or the implementation of existing legislations [76,82]. The aim to become middle income countries shifted the strategic economic focus on macro-economic projects. In particular, in Uganda, the written policy legislations do not include environmental impact assessments. Resulting thereof, monitoring, evaluation and the implementation of climate policies and climate change interventions is not existing. Due to the high amount of international cash flow received, in Kenya, environmental impact assessments are included in economic policies. However, the implementation of these projects shows that the motivation to enforce these assessments is low.

The lack of a clear-cut system of responsibilities is appreciated by (inter)national business actors. It allows them to engage in national decision-making processes and to influence national policies in relation to their interests. Hence, the current institutional framework creates indirect governance systems, that make it easier for them to implement their business priorities. This discrepancy results in complicated linkages with other policy areas, as different actors pull in different directions. In both countries, the politically responsible actors might have engaged in multilateral and bilateral climate

financing opportunities, however, with reluctance. Therefore, they did not get involved in driving climate policies on the ground.

Whereas climate finance should follow a bottom-up approach to be better equipped and better able to adapt to the changes on the ground, climate policies and national decision-making are still implemented via a top-down approach in Kenya and Uganda [58,59,78,90]. The lack of political willingness to allow sub-national participation in climate policies leads to ineffective climate change adaptation processes, and therewith, a non-existent climate financing program. While the institutional frameworks shall provide the opportunity for climate policies and climate financing, thus, enabling an international cash flow from international to domestic institutions, the national governments show their unwillingness to use the received sources. Resulting thereof, the climate change vulnerability of both lake sites is increasing. Furthermore, earlier attempts by the county government to acquire control of the political action space and economic gains have led to a debate as to whether natural resource management should follow administrative boundaries, or should still be managed under hydrological boundaries in Kenya.

Climate change adaptation and sustainable climate policies are more than a mere donor agenda in the study countries. In many countries of the Global South climate, financing is an arena for struggles over authority between the central state and 'twilight institutions', such as sub-national governments and customary institutions. This is especially the case in countries which are both highly vulnerable to being affected by climatic changes, and transitioning from least developed to middle-income countries. In a context where conventional donors have lost clout as a result of the growth in developing economies and increasing foreign investment flows (including climate cash flows) to and across the South, climate financing is used by national governments towards economic projects. Furthermore, international donors do not follow up the usage of their provided cash flow appropriately. In the case of Lake Naivasha, the World Bank Group is supporting the construction of a Cargo Center, which should ease the trade between Kenya and Uganda. Due to the fact that the Cargo Center is constructed in Hell's Gate National Park, the World Bank Group requested an environmental impact assessment of the economic investment undertaken. Independently from the Cargo Center's negative impact on the environment and the wildlife, the construction already started, with the financial support of the World Bank Group. Resulting thereof, the economic developments in these countries are also of upmost importance to international donors, as key selling markets and production facilities.

Decentralizing climate funds can be used as a major component to limit climate change vulnerability, as it can help to ensure that resources reach where they are needed most. However, there is a need to enhance the institutional framework between the national government and sub-national level. While county governments provide a good opportunity to create institutional linkages for devolving funds from the national to the sub-national, there are no clear linkages between national climate financial mechanisms and county ones. Thus, there is a need to create clear structural linkages between the National Climate Finance Unit and the county climate change funds at county treasuries.

## 6. Conclusions

This article has demonstrated that climate financing is dependent on the country's governance architecture, as well as the interplay between national and sub-national institutions, that enable climate financing through international cash flows. The case studies have shown that there is no clear link to the overall nature of the state (e.g., strong or weak). Rather, it seems to be a question of the state's particular historical trajectories and the current conditions and context in which the state finds itself. Climate change thus offers an opportunity for sub-national governments, not only to counter state interventions, but also to set out their own claims to authority and legitimacy vis-à-vis each other and the central state. The establishment of a decentralized political system seems to be a major component of climate financing. It might help to ensure that resources reach where they are needed most. However, there is a need to enhance the institutional connection between the national government and sub-national

level, as this currently remains, particularly in the least developed and developing countries, weak. While county governments provide a good opportunity to create institutional linkages for devolving funds from the international to the sub-national, there are no clear linkages between national climate financial mechanisms and county ones. There is a need to create clear structural linkages between the National Climate Finance Unit and the county climate change funds at county treasuries.

**Funding:** Open Access funding supported and provided by the University of Koblenz-Landau.

**Acknowledgments:** I would like to thank the three external reviewers as well as Axel Marx and Emilie Becault for their helpful comments and suggestions.

**Conflicts of Interest:** The author declares no conflict of interest.

## Appendix A

**Table A1.** Stakeholder Analysis Questionnaire.

| **Which actors are involved in climate policy decision-making or have a general interest in climate governance?** |
| :---: |
| Which actors are on the national level? |
| Which actors are on the sub-national level? |
| Which actors are on the local level? |
| Which actors are on the international level? |
| **Level of Influence and Interests** |
| Which of the named actors do you consider to have high influence on the institutional framework? Which actors have low influence on the institutional framework? |
| Which of the named actors do you consider to have high interest in climate policies? Which actors have low interests in climate policies? |

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
