# Peer review of "New Power Structures and Shifted Governance Agendas Disrupting Climate Change Adaptation Developments in Kenya and Uganda"

_sustainability, doi:10.3390/su12072799_

Round 1

Reviewer 1 Report

This is an interesting contribution that sits within the core theme of the Special Issue of studying the interplay between international factors and domestic institutions in governing adaptation to climate change. It should be included in the Special Issue provided that it makes substantive improvements in a revised version. My main issue is not so much with the substance of the research (I am not really qualified to assess it) but with the way it is presented. A revised version of the paper would need to prune contents more rigorously (or better connect the various arguments) and guide the reader much better.

While I usually start my review reports with a summary of the underlying paper, I had trouble to do so here. The paper at present is unfocused and raises a series of interesting questions, but no coherent story line exists as of yet. What is the actual research question this paper seeks to address? What is the key argument? If that is too big a task, then try to break this down and identify a 'dependent variable' (what exactly is the focus here -- climate policy output, or outcomes like emissions reductions or adaptation benefits?), the key 'independent variable' (level of decentralization? access to international finance? Their interaction?), and what are the control variables (all things that are similar in Uganda and Kenya, for instance colonial legacy, socioeconomic status, geography and vulnerability). I hope that thinking in these terms will help.

I was expecting to learn from this paper how political institutions mediate the effects of interests and influences in the governance of climate policy in two Lake regions in two African states. This would be my faithful summary from reading through the main body, but the abstract does not reflect that. Later in the introduction the paper states its goals rather clearly:
(I) to identify the key characteristics of the institutional framework that shapes natural resource and climate governance
(II) the role and weight of internal decision-making processes and strategic priorities is discussed in shaping the overall climate change adaptability agenda

I am unable to recall the answer to these questions. On goal I, is the key characteristic the level of decentralization that the national state allows? In Kenya, climate governance seemed more devolved (or decentralized? Please clarify as these are two different things) than in Uganda. On goal II, I am not sure I read about internal decision-making processes. The abstracts seems to indicate that investments by national and international companies are the main 'treatment'.

Relatedly, it seems that the paper contradicts itself on some issues. First, it states the puzzle "why climate governance is failing in Kenya and Uganda despite international cash flows?" (which does not even seem puzzling to me because money alone can create perverse incentives, so will not automatically improve governance? See resource curse literature); later it states that "climate finance and climate governance are not working" (line 246) which apparently contradicts the first statement. Because the paper hides behind nominalizations ("failure of climate finance" -- what does this mean? Who fails? How? And why?) these things remain unclear.

The structure does not seem intuitive either. Section 2 should only talk about the theoretical framework. I have been missing the polycentric governance framework by Ostrom, which would be useful here, but the Stakeholder Analysis seems to do a similar job so I am fine with that. However, in its current form, it is incomprehensive. SA is just a framework and as such does not offer theoretical insight into the puzzle stated earlier. On structure, I suggest to have 2.4 as free-standing (with parts of 3.3) theory section, followed by a new section 3 which describes the case selection and research method. Section 4 should present the results, but less descriptively than at present (i.e., more analytical). I could not follow how the authors came up with their influence matrix for the two cases? On which basis were these scores assigned? Also, Table 1 and 2 are overwhelming. They make it impossible to understand policy outcomes based on a set of predictors (which ones? apparently the influence? But has this influence been inferred from outcomes?). In moving forward, I suggest the paper should focus on explaining more carefully the interests of the national government, sub-national government, firms, NGOs, and any other relevant actor (without disaggregating them), and show how they interacted (in the given institutional context) to produce the outcome. Which role did (access to) international finance play? By focusing on just a few actors, specifically the key ones, it would be much easier to have a systematic discussion.

One could even envisage to formulate a theory from a government's perspective only. The abstracts seems to do so: Is there a tradeoff between socioeconomic development and climate policy objectives? And if so, how does climate finance help resolve such tradeoff? This would be a fundamentally important question that didn't quite get covered. Or is the point of the paper that no actor really sits in the driver's seat? In any case, one would need to specify what other actors wanted in the particular cases reviewed.

In terms of empirics, what is the purpose of two case study countries? I was expecting to see some sort of systematic comparison (a la Mill's methods) between Kenya and Uganda to isolate some causal factor, but apparently both cases fit the same pattern (despite small differences in institutional details, see "Since the contradiction between sustainability, environmental awareness and environmental destruction is so great both at Lake Naivasha in Kenya and at Lake Wamala in Uganda, the two lakes serve as case studies for this research article."). I would like the authors to clarify the role of the case studies upfront in the paper.

These thoughts are by no means exhaustive -- but I feel it makes little sense to continue because it is unclear to me what this paper tries to convey. I am fairly open to any kind of argument in a revised version as long as there is one.

Finally, a couple of smaller points. There is a need for language editing, simply to clarify what exactly the authors mean. It would be important to avoid colloquial terms and be precise. What are "semi-influence" and "semi interests"? Either an actor has influence or not? (I don't know, but noone explained). Smaller typos: 'developments' should read 'development' (in the abstract); World Market, Climate Targets, Crowd, Stakeholder Analysis should be in lower case (drop comma that ensues after World Market), countries -> countries' (in abstract). Finally, the paper raises the issue of co-benefits but does not deepen it (l.144-45).

Author Response

Response to Reviewer 1 Comments

Point 1: The paper at present is unfocused and raises a series of interesting questions, but no coherent story line exists as of yet. What is the actual research question this paper seeks to address? What is the key argument? If that is too big a task, then try to break this down and identify a 'dependent variable' (what exactly is the focus here -- climate policy output, or outcomes like emissions reductions or adaptation benefits?), the key 'independent variable' (level of decentralization? access to international finance? Their interaction?), and what are the control variables (all things that are similar in Uganda and Kenya, for instance colonial legacy, socioeconomic status, geography and vulnerability). I hope that thinking in these terms will help.

Response 1: Dear Reviewer, I appreciate your time and comments on my paper. In this section, I would like to respond to our comments about the Introduction and storyline of the paper. Concerning your question, about the research question, Line 118 states now the research question “How do the institutional frameworks shape natural resource and climate governance in Kenya and Uganda?” More so, line 116 to 117 highlight the key variables (DV: climate policy output; IV: level of decentralization (i.e. institutional framework). Furthermore, I re-structured the introduction to accentuate the focus of the paper and its relevance. Accordingly, line 25-58 contain general information about climate change and climate financing in especially developing countries and how the institutional framework can support effective climate financing. Line 59-96 state the relevance of the chosen case studies and draw on existing information. Line 97-114 refer to the relevance of the case selection and highlight the value the article will add to existing studies. Line 116-131 state the aim, research question, objective and methodology of the paper. The remaining lines (132-137) present the structure of the paper.

Point 2: The structure does not seem intuitive either. Section 2 should only talk about the theoretical framework. I have been missing the polycentric governance framework by Ostrom, which would be useful here, but the Stakeholder Analysis seems to do a similar job so I am fine with that. However, in its current form, it is incomprehensive. SA is just a framework and as such does not offer theoretical insight into the puzzle stated earlier. On structure, I suggest to have 2.4 as free-standing (with parts of 3.3) theory section

I could not follow how the authors came up with their influence matrix for the two cases? On which basis were these scores assigned?

Response 2: Dear Reviewer, thank you very much for your comments concerning Section 2. I re-arranged the structure. Thus, the former Section 2.4. is now a free-standing section elaborating on the theoretical framework and its contribution to solve the mentioned puzzle in Section 1. Based on your comment on Ostrom, I included a few introducing sentences and highlighted why Stakeholder Analysis is chosen over Ostrom (line 140-162) and how combination of climate governance and stakeholder analysis makes a theoretical contribution to the paper. The section follows by an outline of governance (line 164-176) and a more specific section on climate governance (line 177-204). The last section of the theoretical section takes a closer look at Stakeholder Analysis. I included a few background information (who developed it à line 208 and why the four-group categorization is chosen à line 227). Furthermore, information is added to allow for a better understanding how scores have been assigned (including a reference to the methodological section).

Point 3: followed by a new section 3 which describes the case selection and research method.

Response 3: Dear Reviewer, thank you very much for your comments regarding the methodology. As indicated in response 2, there is now a new Section 3 looking in particular at the materials and methods used. The methodology section was revised adding more technical details on the interviews. Line 363-365 contains information on how many interviews have been conducted, when and how often. More so, line 375-380 provides information on how results have been derived from the interviews especially regarding the influence-interest matrix. This also helps to better understand how the matrix was constructed.

Point 4: Section 4 should present the results, but less descriptively than at present (i.e., more analytical). I could not follow how the authors came up with their influence matrix for the two cases? On which basis were these scores assigned? Also, Table 1 and 2 are overwhelming. They make it impossible to understand policy outcomes based on a set of predictors (which ones? apparently the influence? But has this influence been inferred from outcomes?).

It would be important to avoid colloquial terms and be precise. What are "semi-influence" and "semi interests"? Either an actor has influence or not? (I don't know, but noone explained)

In Kenya, climate governance seemed more devolved (or decentralized? Please clarify as these are two different things) than in Uganda

Response 4: Dear Reviewer, thank you very much for the comments. To better structure the empirical part of the paper, I included, first, a general introduction to guide the reader through the result section. Section 4.1. (governance overview) and 4.2. (description of stakeholders) is more descriptive whereas the following two are more analytical. In the general introduction (line 390-395) I highlighted that the Table 1 and Table 2 present the most important actors who have both interest and influence on climate policy in the two countries. However, the tables do not give information if the actors have high or low interest. For the analytical part of the Stakeholder Analysis, figure 2 and 3 are included.

I changed the wording for influence and interest to low or high interest, skipping the phrase of semi-interest to emphasize if an actor has influence/interest or not.

Furthermore, I highlighted in Section 4.1. the difference between decentralization and devolution. The official term for introducing decentralized structures in Kenya is devolution. In Uganda, there does not exist an official term, therefore, decentralization is used.

Point 5: In moving forward, I suggest the paper should focus on explaining more carefully the interests of the national government, sub-national government, firms, NGOs, and any other relevant actor (without disaggregating them), and show how they interacted (in the given institutional context) to produce the outcome. Which role did (access to) international finance play? By focusing on just a few actors, specifically the key ones, it would be much easier to have a systematic discussion.

Response 5: Dear Reviewer, thank you for your comments regarding the Section 4.4. (Interaction of interests and influences). After the description of the stakeholder’s interest of climate policy outcome, Section 4.4. focusses on the second research objective: the role and weight of internal decision-making processes and strategic priorities is discussed in shaping the overall climate policy agenda. It discusses the relationship between the interests and the influences of the identified stakeholder groups. The section will show that access to climate finance is determined by the influence the actors have on the institutional framework independently their interests in climate policy. After a short description of the Stakeholder Analysis figure (and how the classification was done line 580-585), all stakeholder groups are briefly touched upon regarding their influences on the institutional framework (line 586-598 national actors; line 599-609 economic actors; line 610-627 sub-national actors; line 628-643 NGOs and locals). The section (line 644-662), furthermore, highlights that other than national and sub-national actors, all others are not necessary to consider much more in detail in the following section (discussion). The section closes with a summary and a small discussion on the interaction of national and sub-national actors to access international climate finance and their impact on the institutional framework. Thus, only these two actor groups will be considered in the following section (Section 5: Discussion) to allow for a better discussion of the introduced research questions and the two research objectives stated.

Point 6: I would like the authors to clarify the role of the case studies upfront in the paper. The abstracts seems to do so: Is there a tradeoff between socioeconomic development and climate policy objectives? And if so, how does climate finance help resolve such tradeoff? This would be a fundamentally important question that didn't quite get covered. Or is the point of the paper that no actor really sits in the driver's seat?

Response 6: Dear Reviewer, thank you very much for the comments on the discussion part and the abstracts. The discussion was restructured and revised to allow for a more systematic analysis of the stated research question and the two research objectives. The first part of the discussion (line 664-676) highlights the general tensions between national and sub-national actors in matters related to climate policy. In the upcoming two parts (line 677-687 and 688-696) the development of the institutional framework in Kenya and in Uganda are discussed independently from each other. The section on Kenya highlights that especially sub-national actors changed their behaviour from an initial excitement about devolution to frustration and later to resignation. Furthermore, political kinship and ethnic affiliations also hamper climate policy outcomes on a sub-national basis. Afterwards, the case of Uganda is discussed. It is highlighted that climate policy is not included in national nor sub-national decision-making processes. This discussion on the institutional framework concludes with a comparison of the institutional frameworks of both countries. Line 706-713 shows how economic actors use the existing institutional frameworks to integrate their economic agendas in the institutional framework and furthermore hamper climate policies even more. Lastly, the discussion concludes (line 714-739) with a general discussion how the institutional framework influences climate policies in Kenya and Uganda and argues that climate policies is not integrated in the institutional framework as nobody is considering it to be an important policy currently. Lastly, line 725-732, show that the results drawn from the two case studies can be transferred to other countries in the Global South.

The abstracts at the beginning of the paper was revised to better fit the entire research article.

Reviewer 2 Report

Referee Report on „New Power Structures and Shifted Governance Agendas Disrupting Climate Change Adaptation Developments in Kenya and Uganda”

Summary: The paper investigates the role of national and subnational governance in mitigation and adaptation processes in the African context, focusing on Kenya and Uganda as case studies. The paper particularly seeks an answer to the question why climate governance fails in both countries despite sufficient international flows. It is an important research question contributing to the emerging literature on the interplay of climate governance and finance.  However, some issues require major revision and/or further clarifications.

Comments:

  • Section 1: This section needs to be more concisely written, describing the research question, the importance of the question, explaining the value-added in the literature, the methodology to answer the question and summarizing the results. More particularly, the author also needs to motivate why Uganda and Kenya are selected for a comparison purpose, rather than any other Sub-Saharan African country. Although in the introduction the dissimilarities between the countries are emphasized as a motivation, neither the following discussions nor the results are linked to those dissimilarities. It is not clear what we learn from the comparison of the two countries.
  • 2.4: The analysis of the question relies on stakeholder interviews and adopts the so-called “Stakeholder Analysis (SA)” in order to take into account the interests and interactions of the stakeholders in a given setting. In qualitative analyses, the use of stakeholder interviews is a common tool and there are several frameworks (in political economics) taking into consideration stakeholder interests, objectives, and their interactions. The author should convince the reader why she prefers the SA framework over others.
  • 2.4: There needs further clarifications about the technical details of the SA framework. Who developed the SA framework? Why does the author uses a 4-group categorization? Is it an ad hoc categorization or established within this framework? Do these categories vary across country contexts? In explaining the categories, it would be better to exemplify them as well.
  • 2.4: The discussion of this subsection is too general. It would be more useful to put the discussion in the contexts of Kenya and Uganda.
  • Section 3.1: For the understanding of international readers, it would be useful to discuss Kenya and Uganda statistics in comparison to some European countries and/or USA.
  • Section 3.2: The discussion of the results should be rewritten, particularly after the description of Table 1 and Table 2. It is difficult to follow to which country the presented results apply. Are these results coming from the interviews? How does the author derive the results discussed here?
  • Section 3.3: My last two comments above also apply to this subsection: how to derive these conclusions?
  • Section 3.3: The author should have made it clear earlier that the categorization is made after observing influence-interest matrix. For example, the context players differ in two countries and this is decided based on the matrixes, I suppose. It should be also clarified how to construct the influence-interest matrix.
  • Section 3: The literature discussion spares a large room for climate finance, and the reader expects so see more result on climate finance. However, it is not the case for the results. So I would recommend a proportional and parallel discussion between sections.
  • Section 3: More technical details are to be provided regarding the interviews; for example, the number of interviews and the composition of interviewees in order to understand the representativeness of the sample. In a qualitative analysis, it is also important to know how the results are derived from the interviews. The author could also provide the interview questions in the appendix, which would help to understand how to derive the results. 

Author Response

Response to Reviewer 2 Comments

  • Point 1: Section 1: This section needs to be more concisely written, describing the research question, the importance of the question, explaining the value-added in the literature, the methodology to answer the question and summarizing the results. More particularly, the author also needs to motivate why Uganda and Kenya are selected for a comparison purpose, rather than any other Sub-Saharan African country. Although in the introduction the dissimilarities between the countries are emphasized as a motivation, neither the following discussions nor the results are linked to those dissimilarities. It is not clear what we learn from the comparison of the two countries.
  •  

Response 1: Dear Reviewer, I appreciate your time and comments on my paper. In this section, I would like to respond to our comments about Section 1 especially concerning the outline and highlighting the importance of the paper. Concerning your question, about the research question, Line 118 states now the research question “How do the institutional frameworks shape natural resource and climate governance in Kenya and Uganda?” More so, line 116 to 117 highlight the key variables (DV: climate policy output; IV: level of decentralization (i.e. institutional framework). Furthermore, as you correctly indicated, I re-structured the introduction to accentuate the focus of the paper and its relevance. Accordingly, line 25-58 contain general information about climate change and climate financing in especially developing countries and how the institutional framework can support effective climate financing. Line 59-96 state the relevance of the chosen case studies and draw on existing information. Line 97-114 refer to the relevance of the case selection and highlight the value the article will add to existing studies. Line 116-131 state the aim, research question, objective and methodology of the paper. The remaining lines (132-137) present the structure of the paper.

The remaining paper will also focus on the dissimilarities of the two case studies, as introduced in section 1, better. However, I will not respond to this aspect in this section of the response. I will refer to it later.

  • Point 2: The analysis of the question relies on stakeholder interviews and adopts the so-called “Stakeholder Analysis (SA)” in order to take into account the interests and interactions of the stakeholders in a given setting. In qualitative analyses, the use of stakeholder interviews is a common tool and there are several frameworks (in political economics) taking into consideration stakeholder interests, objectives, and their interactions. The author should convince the reader why she prefers the SA framework over others.
  • 4: There needs further clarifications about the technical details of the SA framework. Who developed the SA framework? Why does the author uses a 4-group categorization? Is it an ad hoccategorization or established within this framework? Do these categories vary across country contexts? In explaining the categories, it would be better to exemplify them as well.
  • The discussion of this subsection is too general. It would be more useful to put the discussion in the contexts of Kenya and Uganda

Response 2: Dear Reviewer, thank you very much for your comments concerning Section 2.4. As the theoretical explanations are key to the puzzle and the first research objective introduced in section 1, I decided to write a free-standing Section 2 out of the former section 2.4. The nee section 2 elaborats on the theoretical framework and its contribution to solve the mentioned puzzle in Section 1. Based on your comment concerning why the SA framework was chosen, I included introducing sentences and highlighted why Stakeholder Analysis is chosen over Ostrom (line 140-162) and how combination of climate governance and stakeholder analysis makes a theoretical contribution to the paper. The section follows by an outline of governance (line 164-176) and a more specific section on climate governance (line 177-204). The last section of the theoretical section takes a closer look at Stakeholder Analysis. I included a few background information (who developed it à line 208 and why the four-group categorization is chosen à line 227). The four-group categorization was established in the core concept and has been used since then as one tool to analyse the actors and their influences independently from the decision-making process, context or country used. Furthermore, information is added to allow for a better understanding how scores have been assigned (including a reference to the methodological section). Concerning your final point (too general discussion): this part has been excluded from the new section 2 and was modified in section 4 to have a more analytical discussion of climate financing in Kenya and Uganda.

The remaining sections from the former section 2 (literature review, research area and method) have been moved to a new section 3.

Point 3: Section 3.3: The author should have made it clear earlier that the categorization is made after observing influence-interest matrix. For example, the context players differ in two countries and this is decided based on the matrixes, I suppose. It should be also clarified how to construct the influence-interest matrix.

  • Section 3: The literature discussion spares a large room for climate finance, and the reader expects so see more result on climate finance. However, it is not the case for the results. So I would recommend a proportional and parallel discussion between sections.
  • Section 3: More technical details are to be provided regarding the interviews; for example, the number of interviews and the composition of interviewees in order to understand the representativeness of the sample. In a qualitative analysis, it is also important to know how the results are derived from the interviews. The author could also provide the interview questions in the appendix, which would help to understand how to derive the results. 

Response 3: Dear Reviewer, thank you very much for your comments regarding the literature review and the methodology. As indicated in response 2, there is now a new Section 3 looking in particular at the materials and methods used. Concerning your points about the literature review, in the following discussion (Section 4 and 5), an emphasise is given on the three main points raised during the literature review to highlight the results on climate finance. These points include: to address the domestic development objectives parallel to the economic investments undertaken; to address powerful interest groups and how they undermine the climate policy outcomes, and third discussing the three objectives highlighted in line 298-304.

Furthermore, the methodology was revised adding more technical details on the interviews. Line 363-365 contains information on how many interviews have been conducted, when and how often. More so, line 375-380 provides information on how results have been derived from the interviews especially regarding the influence-interest matrix. This also helps to better understand how the matrix was constructed.

Point 4: Section 3.1: For the understanding of international readers, it would be useful to discuss Kenya and Uganda statistics in comparison to some European countries and/or USA.

Section 3.2: The discussion of the results should be rewritten, particularly after the description of Table 1 and Table 2.

Response 4: Dear Reviewer, thank you very much for the comments. To better structure the empirical part of the paper, I included, first, a general introduction to guide the reader through the result section. Section 4.1. (governance overview) and 4.2. (description of stakeholders) is more descriptive whereas the following two are more analytical. At the end of section 4.1. information has been added allowing for a better comparison of the climate spending of Kenya and Uganda to other countries, e.g. Germany, France or the US. However, it is difficult to compare them, as each country is having different ministries which are responsible for the climate sector. In some countries, climate spending is under the sector for environmental affairs and sustainability (France), Germany, e.g. names it under the sector for water and environmental spending, whereas the US do not have a sector per se and just count the money spend for environmental projects. As a result, I am hesitating to include this comparison as a reliable comparison is not possible.

In the general introduction (line 390-395) I highlighted that the Table 1 and Table 2 present the most important actors who have both interest and influence on climate policy in the two countries. However, the tables do not give information if the actors have high or low interest. For the analytical part of the Stakeholder Analysis, figure 2 and 3 are included.

I changed the wording for influence and interest to low or high interest, skipping the phrase of semi-interest to emphasize if an actor has influence/interest or not.

  • Point 5: It is difficult to follow to which country the presented results apply. Are these results coming from the interviews? How does the author derive the results discussed here?

Response 5: Dear Reviewer, thank you for your comments regarding the result section and how the results have been derived from the interviews. After the description of the stakeholder’s interest of climate policy outcome, Section 4.4. focusses on the second research objective: the role and weight of internal decision-making processes and strategic priorities is discussed in shaping the overall climate policy agenda. It discusses the relationship between the interests and the influences of the identified stakeholder groups. The section will show that access to climate finance is determined by the influence the actors have on the institutional framework independently their interests in climate policy. After a short description of the Stakeholder Analysis figure (and how the classification was done based on the interviews line 580-585), all stakeholder groups are briefly touched upon regarding their influences on the institutional framework (line 586-598 national actors; line 599-609 economic actors; line 610-627 sub-national actors; line 628-643 NGOs and locals). The section (line 644-662), furthermore, highlights that other than national and sub-national actors, all others are not necessary to consider much more in detail in the following section (discussion). The section closes with a summary and a small discussion on the interaction of national and sub-national actors to access international climate finance and their impact on the institutional framework. Thus, only these two actor groups will be considered in the following section (Section 5: Discussion) to allow for a better discussion of the introduced research questions and the two research objectives stated.

  • Point 6: Section 3.3: My last two comments above also apply to this subsection: how to derive these conclusions?

Response 6: Dear Reviewer, thank you very much for the comments on the discussion part and the abstracts. The discussion was restructured and revised to allow for a more systematic analysis of the stated research question and the two research objectives. The first part of the discussion (line 664-676) highlights the general tensions between national and sub-national actors in matters related to climate policy. In the upcoming two parts (line 677-687 and 688-696) the development of the institutional framework in Kenya and in Uganda are discussed independently from each other. The section on Kenya highlights that especially sub-national actors changed their behaviour from an initial excitement about devolution to frustration and later to resignation. Furthermore, political kinship and ethnic affiliations also hamper climate policy outcomes on a sub-national basis. Afterwards, the case of Uganda is discussed. It is highlighted that climate policy is not included in national nor sub-national decision-making processes. This discussion on the institutional framework concludes with a comparison of the institutional frameworks of both countries. Line 706-713 shows how economic actors use the existing institutional frameworks to integrate their economic agendas in the institutional framework and furthermore hamper climate policies even more. Lastly, the discussion concludes (line 714-739) with a general discussion how the institutional framework influences climate policies in Kenya and Uganda and argues that climate policies is not integrated in the institutional framework as nobody is considering it to be an important policy currently. Lastly, line 725-732, show that the results drawn from the two case studies can be transferred to other countries in the Global South.

The abstracts at the beginning of the paper was revised to better fit the entire research article.

Reviewer 3 Report

After incorporating the comments below, the manuscript is well suited for publication in Sustainability.

Introduction / Literature review:

Given the topic of Impacts of capital flows on the climate change agenda in developing countries, I was surprised to see no mention of foreign aid (and the impact of aid on sustainable development) provided by bilateral and multilateral donors. The topic was pioneered by Hicks and his colleagues (see Hicks, R.L., Parks, C.B., Timmons, R.J., Tierney, M.J., 2008. Greening Aid? Understanding the Environmental Impact of Development Assistance. Oxford University Press, New York) and it was further elaborated for instance by Oprsal et al. (Opršal, Z., HarmáÄŤek, J. 2019. Is foreign aid responsive to environmental needs and performance of developing countries? Case study of the Czech Republic. Sustainability, 11 (2), 401). Inclusion of discussion on environmental aid, but also dirty aid ("sensu Hicks", vis-á-vis other capital flows) in Introduction (or Literature Review) would increase the information value of the article.

Another missing theoretical literature concerns the issue of sustainable management of Commons elaborated by Elinor Ostrom. The “governance of environmental and climate issues” (line 279) is indeed governance of Commons. This issue (at an international level, which however is not the main subject of the article) includes the concepts of "prisoner's dilemma" and "Free Riders". However, for the community level (which is implicitly the subject of the manuscript), the prerequisites of efficient management of natural resources are analyzed in detail by Elinor Ostrom (see, for instance, Ostrom, E. (1992). Crafting Institutions for Self-Governing Irrigation Systems. San Francisco, CA: Institute for Contemporary Studies or Ostrom, E. (ed.) (2002). The Drama of the Commons. Washington, DC: National Academy Press.).

Methodology and Data Collection

In this section, I am missing information on how many stakeholder interviews have been conducted and at what year (or season) they have taken place. It might be sufficient to clarify whether each of the stakeholders listed in Tables 1 and 2 included one or more interviews.

More:

Include a simple overview map of the areas covered (in the context of Kenya and Uganda / or East Africa - not all readers have good situational awareness of these areas).

Author Response

Response to Reviewer 3 Comments

Point 1: Another missing theoretical literature concerns the issue of sustainable management of Commons elaborated by Elinor Ostrom. The “governance of environmental and climate issues” (line 279) is indeed governance of Commons. This issue (at an international level, which however is not the main subject of the article) includes the concepts of "prisoner's dilemma" and "Free Riders". However, for the community level (which is implicitly the subject of the manuscript), the prerequisites of efficient management of natural resources are analyzed in detail by Elinor Ostrom (see, for instance, Ostrom, E. (1992). Crafting Institutions for Self-Governing Irrigation Systems. San Francisco, CA: Institute for Contemporary Studies or Ostrom, E. (ed.) (2002). The Drama of the Commons. Washington, DC: National Academy Press.).

Response 1: Dear Reviewer, I appreciate your time and comments on my paper. Dear Reviewer, thank you very much for your comments concerning the theoretical section of the paper. As the theoretical explanations are key to the puzzle and the first research objective introduced in section 1, I decided to write a free-standing Section 2 out of the former section 2.4. The new section 2 elaborates on the theoretical framework and its contribution to solve the mentioned puzzle in Section 1. Based on your comment concerning why the SA framework was chosen, I included introducing sentences and highlighted why Stakeholder Analysis is chosen over Ostrom (line 140-162) and how combination of climate governance and stakeholder analysis makes a theoretical contribution to the paper. The section follows by an outline of governance (line 164-176) and a more specific section on climate governance (line 177-204). The last section of the theoretical section takes a closer look at Stakeholder Analysis. I included a few background information (who developed it à line 208 and why the four-group categorization is chosen à line 227). The four-group categorization was established in the core concept and has been used since then as one tool to analyse the actors and their influences independently from the decision-making process, context or country used. Furthermore, information is added to allow for a better understanding how scores have been assigned (including a reference to the methodological section). Concerning your final point (too general discussion): this part has been excluded from the new section 2 and was modified in section 4 to have a more analytical discussion of climate financing in Kenya and Uganda.

The remaining sections from the former section 2 (literature review, research area and method) have been moved to a new section 3.

Point 2: In this section, I am missing information on how many stakeholder interviews have been conducted and at what year (or season) they have taken place. It might be sufficient to clarify whether each of the stakeholders listed in Tables 1 and 2 included one or more interviews. Include a simple overview map of the areas covered (in the context of Kenya and Uganda / or East Africa - not all readers have good situational awareness of these areas).

Response 2: Dear Reviewer, thank you very much for your comments regarding the methodology. As indicated in response 1, there is now a new Section 3 looking in particular at the materials and methods used. The methodology was revised adding more technical details on the interviews. Line 363-365 contains information on how many interviews have been conducted, when and how often. More so, line 375-380 provides information on how results have been derived from the interviews especially regarding the influence-interest matrix. This also helps to better understand how the matrix was constructed. Furthermore, a map on the location of the lakes and the areas was included as well.

Round 2

Reviewer 1 Report

Much improved manuscript, along the following dimensions:

* elements of stakeholder analysis

* research design and research methodology

* stakeholder analysis grid and better identification of key actors' interests

One point for consideration is this. Why is it that international donors cannot exert more influence over national policy makers in that they could ask for more appropriate projects being chosen that help the enviroment? Is this part of a goal conflict where economic development is also important to international donors (as well as 'ownership' by country elites?). I was just surprised that donors would not exert more control here.

Some smaller remaining points:

* remove xxx in paper and annex

* cut down paper to conform to word limits
-- 200 words for abstract
-- not sure about the entire paper and defer to the editor on this but I think it would be important for consistency if all papers were similar in length

* resulting thereof --> as a result

* I would refer to "Subjects" (with quotation marks), not Subjects (without), when introducing the stakeholder grid, to facilitate readability, similar for other terms therein

Author Response

Response to Reviewer 1 Comments

Point 1: Why is it that international donors cannot exert more influence over national policy makers in that they could ask for more appropriate projects being chosen that help the enviroment? Is this part of a goal conflict where economic development is also important to international donors (as well as 'ownership' by country elites?). I was just surprised that donors would not exert more control here.

Response 1: Dear Reviewer, I appreciate again your time and comments on my paper. I included a few sentences on the ambivalent role of international donors in climate financing in developing countries (Line 757-764). An example of an economic project funded by the World Bank Group was used to highlight the role international donors play and to point out that international donors are also interested in the economic developments on the ground.

Point 2: Some smaller remaining points:

Response 2: Dear Reviewer, I appreciate your comments. The supplementary file as well as the references of the tables and figures are now aligned with each other. Furthermore, the abstract is cut down to 200 words. In terms of readability, a few changes have been made.

Reviewer 2 Report

The text has improved substantially upon the revision. Nevertheless, I still have some minor issues to be fixed in the paper. 

1- A brief summary of the findings from the current analysis is missing in the Introduction section. 

2- Ostrom is the leading scholar in the development of a theoretical framework. However, there are subsequent seminal works to be mentioned as well. This second section still needs improvement/restructuring in a way to mention the existing theoretical frameworks (including Ostrom's, but more concisely) to analyze stakeholder interviews and explain the pros and cons of the usage of the named frameworks, then conclude why the author prefers the SA framework over others. 

3- The discussion of the results also needs further improvement because it is still unclear how the author technically derives results from the stakeholder interviews. The current revision already gives insights about the influence-interest matrix and a detailed discussion on results. But, the way that the author derived results from interviews is not well described. 

4- In relation to the third comment, I still believe providing the interview questions, maybe in the appendix, would be useful (I couldn't find them in the supplementary documentation either). 

5- The text requires some language editing. Also, in line 224, "xxx2 ought to be replaced. 

Author Response

Response to Reviewer 2 Comments

Point 1: A brief summary of the findings from the current analysis is missing in the Introduction section. 

Response 1: Dear Reviewer, I appreciate again your time and comments on my paper. I included two sentences which summarize the core findings of the presented analysis in line 130 to 134. Afterwards, the structure of the paper is introduced.

Point 2: Ostrom is the leading scholar in the development of a theoretical framework. However, there are subsequent seminal works to be mentioned as well. This second section still needs improvement/restructuring in a way to mention the existing theoretical frameworks (including Ostrom's, but more concisely) to analyze stakeholder interviews and explain the pros and cons of the usage of the named frameworks, then conclude why the author prefers the SA framework over others. 

Response 2: Dear Reviewer, thank you very much for your comments in matters related to the theoretical framework. I added some more information about Ostrom’s framework and used this information to the same time to challenge the concept and to show why it is not useful for the existing paper (line 154-165). In general, I focussed on three critiques: (I) Ostrom fails to fully grasp the importance of macro level actors; (II) the concept fails to consider the importance of social relationships, and (III) fails to take into consideration the importance of power relationships and the inequalities potentially integral to them. A discussion of other relevant concepts which discuss various stakeholder relationships have not been included in detail to avoid an excess of word count.

Point 3: - The discussion of the results also needs further improvement because it is still unclear how the author technically derives results from the stakeholder interviews. The current revision already gives insights about the influence-interest matrix and a detailed discussion on results. But, the way that the author derived results from interviews is not well described. 

4- In relation to the third comment, I still believe providing the interview questions, maybe in the appendix, would be useful (I couldn't find them in the supplementary documentation either). 

Response 3: Dear Reviewer, thank you very much for your comments concerning the result section. At the beginning of section 4 (results) I included a few sentences on how I derived the information from the conducted interviews. In particular I highlighted the questions concerning the interest-influence matrix. (line 411-419). There is some reluctance to provide the interview questions as the situation especially at the lake sites have partly been tense and interviewees did not want to take part in formal interviews as they have been afraid of arrests or other punishments. Resulting thereof, the methodology section entails information on the interview guideline and how it was structured along the broader themes (line 392). For now, the part of the questionnaire is provided which covered the questions on the specific information retrieved for the stakeholder analysis and the interest-influence matrix (see supplementary file). However, one could agree to provide the questionnaire upon request from readers.

Point 4: - The text requires some language editing. Also, in line 224, "xxx2 ought to be replaced. 

Response 4: Dear Reviewer, thank you very much for the comments. Some minor language editing took place also to facilitate the readability of the paper.
